# Reagent-Controlled Highly Stereoselective Difluoromethylation: Efficient Access to Chiral α-Difluoromethylamines from Ketimines

**DOI:** 10.3390/molecules27207076

**Published:** 2022-10-20

**Authors:** Qinghe Liu, Taige Kong, Chuanfa Ni, Jinbo Hu

**Affiliations:** Key Laboratory of Organofluorine Chemistry, Center for Excellence in Molecular Synthesis, Shanghai Institute of Organic Chemistry, University of Chinese Academy of Sciences, Chinese Academy of Sciences, 345 Ling-Ling Road, Shanghai 200032, China

**Keywords:** difluoromethylation, sulfoximine, nucleophilic addition, ketimine, fluorine

## Abstract

A reagent-controlled highly stereoselective reaction between (S)-difluoromethyl phenyl sulfoximine **1** and imines is reported, and this synthetic method provides a variety of enantiomerically enriched α-difluoromethyl amines. The main pros of this approach include high efficiency, high stereoselectivity, and a broad substrate scope, which is probably achieved through a non-chelating transition state.

## 1. Introduction

The difluoromethyl group (CF_2_H) is one of privileged fluoroalkyl groups which has attracted increasing interest due to its unique biochemical properties [1,2,3,4]. For example, it has strong lipophilicity and has been proven to be an isostere of OH and SH. At the same time, the hydrogen atom in the CF_2_H group can serve as a hydrogen bond donor, thus difluoromethyl analogs of the biologically active molecules have the potential to be much more effective drugs compared to its parent molecules. Especially, difluoromethyl compounds have better biological activity than their corresponding trifluoromethyl compounds in some cases [5,6]. Given the above-mentioned properties, α-difluoromethyl amines have been successfully used in the antagonist or inhibitor molecular design. For instance, an NPY antagonist with high Y1 activity and high selectivity for subtype receptors [7] and a drug candidate as a thrombin inhibitor [8] were shown in the Figure 1A.

It is of high importance to develop synthetic methods of chiral α-difluoromethyl amines for the pharmaceutical and biological chemistry, given the fact that the potential dangers of racemic drugs have been documented [9]. Thus, it has attracted many efforts to attain the chiral α-difluoromethyl amines, and they can be divided into three aspects according to the reaction type: (a) chemical resolution [10], (b) asymmetric hydrogenation [11,12], and (c) stereoselective Mannich addition reaction [13] (shown in Figure 1B). After the comparison of the above methods, it was found that the stereoselective addition reaction based on the imine starting material has several advantages: it can not only obtain higher stereoselectivity, but also has a wide range of substrate scope, which can be applicable to both α-monosubstituted difluoromethyl amine and α,α-disubstituted difluoromethyl amine. For instance, Hu group reported the stereoselective addition reaction between *tert*-butyl sulfinyl protected imines and phenyl difluoromethyl sulfone or TMSCF_2_H to generate the enantiomerically enriched α-difluoromethyl amine [14,15] (shown in Figure 1C(i)), which served as an example of substrate-controlled Mannich addition. Recently, we have been devoted to the development of nucleophilic fluoroalkylation by using fluoroalkyl sulfoximine reagents [16,17,18,19,20]. In this context, we were interested in developing a reagent-controlled stereoselective Mannich reaction instead of a substrate-controlled version [21,22,23]. Herein, we report the first (S)-phenyl difluoromethyl sulfoximine (**1**)-enabled highly stereoselective difluoromethylation of imines, affording synthetically valuable chiral α-difluoromethyl amines (shown in Figure 1C(ii)).

## 2. Results

We started our study by examining the (S)-phenyl difluoromethyl sulfoximine **1** and imine **4a** as reaction partners. After a careful variation of reaction parameters, we identified the suitable reaction conditions in which a mixture of sulfoximine **1** (1.0 equiv.), imine **4a** (1.5 equiv.), and methyl lithium (1.2 equiv.) in THF (0.05 M) afforded **5a** in 38% yield with 99/1 dr (Entry 1, Table 1). Further screening revealed that n-butyl lithium is also feasible, which afforded **5a** in 35% yield with 99/1 dr (Entry 2, Table 1). However, sodium bis(trimethylsilyl)amide was not suitable for this reaction (Entry 3, Table 1). Various solvents were evaluated, and THF was found to be an optimal solvent (Entries 4–6, Table 1). However, when HMPA was added, the yield was significantly reduced but with 99/1 dr (Entry 7, Table 1). When the ratio of **1**/**4a**/MeLi was changed to 1/2/2.8, the yield could be increased to 77% and dr 99/1 (Entry 8, Table 1). However, when *N*-Ts and *N*-SPh-**4a** were used, the yield decreased (Entries 9 and 10, Table 1). Further optimization showed that when the concentration and temperature decreased, it could afford the desired product in 90% yield and 99/1 dr (Entries 11 and 12, Table 1).

Then, we examined the substrate scope of the reaction (Figure 1). Reactions with various imines can afford **5a–****m** in high yields (64–99%) and high diastereoselectivity (dr 95/5–99/1). The halo-substituted substrates were tested, and it can afford **5c** (72% yield, dr 97/3) and **5d** (84% yield, dr 95/5). The substituents such as methyl and isopropyl could be tolerated and **5e** (93% yield, dr 99/1) and **5f** (97% yield, dr 99/1) were obtained. This reaction is not sensitive to the position of the substituent on the aromatic ring, and **5g** (99% yield, dr 99/1), **5h** (90% yield, dr 99/1) and **5i** (88% yield, dr 99/1) were afforded. 3,4-Disubstituted aryl ketimine was also tolerated and **5j** (97% yield, dr 99/1) was obtained. When an aryl ethyl ketimine was used, **5k** (95% yield, dr 99/1) was generated. The cyclic imine **4l** was also tolerated with the reaction. In addition, the heteroaromatic ring such as the one in the furyl group can afford the desired product **5****m** (81% yield, dr 99/1).

Although **5i**, **5j** and **5k** were parallel with the previous preliminary results [20], the process for the corresponding HCF_2_-products is vague. To obtain the difluoromethylation products, **5b** could undergo deprotection of the silyl group with aqueous acid to yield NH-**6b** in full conversion. The absolute configuration of **6b** was reported by our group [20], and those of the others were assigned by analog. The process of a reductive alkyl C-S bond cleavage with magnesium and an N-S bond cleavage with triflic acid could afford **7b** in 69% overall yield (Figure 2), which implied the products could be modified diversely, and provided the possibility of accessing chiral amine derivatives, especially those molecules with bioactivities.

On this basis, we are highly interested in what the rationalization of the high diastereoselectivity is. Due to the addition of HMPA not influencing the diastereoselectivity of the difluoromethylation of **4a** with (S)-**1** (Entry 6, Table 1), we proposed that the cation might not participate in the transition state. In addition, it is worth noting that it is different from the reactions of lithiated phenyl monofluoromethyl sulfoximine and imines. Two possible non-chelating transition states TS-1 and TS-2 were envisaged in Figure 3a. Since the repulsive interactions of Ph-Ph in TS-2 are much stronger than those of Ph-CH_3_ in TS-1, TS-1 is the more favorable transition state. In addition, the possible kinetic interpretation of the reaction with enamidation substrates was proposed [24,25]. The nucleophilic addition of monofluoromethyl phenyl sulfoximine to ketimines requires the preproduction of PhSO(NTBS)CHF^−^ [20,26], while the version of difluoromethyl phenyl sulfoximine was achieved in high yield and stereoselectivity by in situ production of PhSO(NTBS)CF_2_^−^ in the presence of strong bases. We analyzed the possible reaction process in the system, and it was summarized in Figure 3b. The production rate of PhSO(NTBS)CF_2_^−^ and its nucleophilic addition rate to ketimine, namely k1 and k2, are rather critical. When k1 and k2 are much larger than k3 and k4, the enamidation of ketimine and the side reaction of methyllithium addition to ketimine could be avoided, which can ensure the high efficiency between difluoromethyl phenyl sulfoximine and ketimines.

## 3. Materials and Methods

### 3.1. General Information

Unless otherwise mentioned, solvents and reagents were purchased from commercial sources and used as received. The solvents CH_2_Cl_2_, CH_3_CN, DMF, and HMPA were distilled from CaH_2_; THF, PhCH_3_, and Et_2_O was distilled over sodium before being used. ^1^H, ^13^C and ^19^F NMR spectra were recorded on a 500 MHz, 400 MHz or 300 MHz NMR spectrometer. ^1^H NMR chemical shifts were determined relative to internal (CH_3_)_4_Si (TMS) at *δ* 0.0 or to the signal of the residual solvent peak: CHCl_3_ in CDCl_3_: *δ* 7.26. ^13^C NMR chemical shifts were determined relative to internal TMS at *δ* 0.0. For the isolated compounds, ^19^F NMR chemical shifts were determined relative to CFCl_3_ at *δ* 0.0. Data for ^1^H, ^13^C and ^19^F NMR were recorded as follows: chemical shift (*δ*, ppm), multiplicity (s = singlet, d = doublet, t = triplet, m = multiplet, q = quartet, br = broad). Coupling constants are reported in hertz (Hz). MS (EI) was obtained on a HP5973N mass spectrometer. HRMS (EI) were recorded on a SATURN 2000 mass spectrometer, HRMS (DART) were obtained on an AGILENT1100 mass spectrometer (Shanghai, China), and HRMS (DART-LTQ FTICR) were recorded on a FTMS-7 mass spectrometer (Shanghai, China).

### 3.2. General Procedure

Under N_2_ atmosphere, to a solution sulfoximine (*S*)-**1** (0.2 mmol, 1.0 equiv.) and **4** (0.4 mmol, 2.0 equiv.) in THF (8.0 mL), MeLi was added (1.6 M in Et_2_O, 0.56 mmol, 2.8 equiv.) slowly at −98 °C. After 30 min, the reaction was quenched with aqueous saturated ammonium chloride (4 mL), followed by extraction with ethyl acetate (3 × 10 mL). The organic phase was washed with brine and then dried over anhydrous MgSO_4_. After the solution was filtered and the solvent was evaporated under vacuum, the residue was subjected to silica gel chromatography to give the major diastereoisomer **5** using petroleum ether/ethyl acetate as eluent.

## 4. Conclusions

In conclusion, we reported the unprecedented stereoselective nucleophilic difluoromethylation of ketimines using chiral difluoromethyl phenyl sulfoximine. The reagent-controlled highly stereoselective reaction features high efficiency and a broad substrate scope. The reductive cleavage of alkyl C-S bond proved that it could serve as a good access to α-difluoromethyl amines. The possible transition states and kinetic interpretation of the reaction were also demonstrated. Not only does our work provide a valuable synthetic tool and new insights into the intriguing reactivity of sulfoximines, but it also serves as a basis for the further development of chiral fluorinated amines.

## Data Availability

The data presented in this study are available on request from the corresponding author.

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
