# Peer review of "Reagent-Controlled Highly Stereoselective Difluoromethylation: Efficient Access to Chiral α-Difluoromethylamines from Ketimines"

_molecules, 2022, doi:10.3390/molecules27207076_

Round 1

Reviewer 1 Report

Comments on molecules-1945254-peer-review-v1

Recommendation

Accepted after minor revision

Comments to the Author:

In this manuscript, Hu and co-workers reported the stereoselective difluoromethylation of imines employing (S)-difluoromethyl phenyl sulfoximine as difluoromethyl reagent. Although the authors have extensively studied the difluoromethylation of imines with fluoroalkyl sulfoximine reagents before, the current reaction can be seen as an expansion reaction. In addition, the title reaction proceeds well and shows a good functional group tolerance to provide numerous enantiomerically enriched α-difluoromethyl amines.

Moreover, the manuscript is well written and the SI is in good shape. In conclusion, I recommend the acceptance of this manuscript for publication in the Molecules after addressing the following points and checking whole manuscript.

(1)   In scheme 1, the protecting group of imines is limited as only Bus substituent is suitable for this type of reactions. Thus, authors should add a couple of examples for comparision, making the current approach more attractive.

(2)   Page 3, When yields were determined by NMR analysis (e.g., Table 1), the internal standard used for analysis should be specified.

Format:

(1)   Page 1, Line 32, Ref 9, The reference number should be placed before the period (e.g. ref. [16-20], [20], [24-25], [20,26] )

(2)   Page 3, Line 65, Number of compounds (e.g. 1a, 4a) should be bolded.

(3)   Page 6, Line 138, ‘Et2O’ should be ‘Et2O’.

(4)   Page 6, Line 139, ‘-98oC’ should be ‘-98 oC’.

(5)   Page 7, Line 192, The ‘Curr. Top. Med. Chem.’ should be in italics.

(6)   Page 7, Line 209, ‘National Science Review’ should be ‘Nat. Sci. Rev.’.

Typos:

(1)   Page 6, ‘ketimiens’ should be ‘ketimines’.

Author Response

 -- Reviewer 1, comment #1: In scheme 1, the protecting group of imines is limited as only Bus substituent is suitable for this type of reactions. Thus, authors should add a couple of examples for comparision, making the current approach more attractive.

Our response: Thank you for the comment. We have added two cases with different protecting group for comparison (N-Ts-4a and N-SPh-4a). And the yields decreased. We have added the results in Table 1 entries 9 and 10.

--Reviewer 1, comment #2: When yields were determined by NMR analysis (e.g., Table 1), the internal standard used for analysis should be specified.

Our response: Thank you for the comment. We have added “…using the PhCF3 as the internal standard” in the subscript b of Table 1

--Reviewer 1, comment #3: Format:

(1)   Page 1, Line 32, Ref 9, The reference number should be placed before the period (e.g.  ref. [16-20], [20], [24-25], [20,26] )

(2)   Page 3, Line 65, Number of compounds (e.g. 1a, 4a) should be bolded.

(3)   Page 6, Line 138, ‘Et2O’ should be ‘Et2O’.

(4)   Page 6, Line 139, ‘-98oC’ should be ‘-98 oC’.

(5)   Page 7, Line 192, The ‘Curr. Top. Med. Chem.’ should be in italics.

(6)   Page 7, Line 209, ‘National Science Review’ should be ‘Nat. Sci. Rev.’.

Our response: All of the format issues have been corrected.

--Reviewer 1, comment #4: Typos:

(1)   Page 6, ‘ketimiens’ should be ‘ketimines’.

Our response: Thank you for the comment. We have corrected it.

Reviewer 2 Report

The paper is written very well, it is clear and concise. The authors explain very well in this paper that the difluoromethyl group (CF2H) is of great interest in the area of medicinal chemistry, fluorine-containing compounds, especially those containing difluoromethylene (CF2) groups, are very useful in agrochemicals, pharmaceuticals, and life science.

I appreciate authors for screening and optimization in different solvents. I have few comments included for the betterment of the paper and please can authors comments on below points.

1.     Can authors comment on if NaH use as base in the reaction instead of nBuLi or MeLi.

2.     Substituted cyclic ketimine can tolerate in this reaction.

3. Can authors observe in the reaction products any nucleophilic addition product of (S)-phenyl difluoromethyl sulfoximine (1), with its PhSO(NTBS)CHFÍž.

Author Response

 -- Reviewer 2, comment #1: Can authors comment on if NaH use as base in the reaction instead of nBuLi or MeLi.

Our response: Thank you for the comment. Considering the readily deprotonation of (S)-phenyl difluoromethyl sulfoximine in low temperature, we selected NaHMDS as the base to generate the corresponding anion. And only 3% NMR yield was observed. We have added the result in Table 1 entry 3.

--Reviewer 2, comment #2: Substituted cyclic ketimine can tolerate in this reaction.

Our response: Thank you for the comment. We tried one case for the cyclic ketimine, and it worked well. And we added the result in Scheme 1.

--Reviewer 2, comment #3: Can authors observe in the reaction products any nucleophilic addition product of (S)-phenyl difluoromethyl sulfoximine (1), with its PhSO(NTBS)CHF- ?

Our response: The products could be generated when (S)-phenyl difluoromethyl sulfoximine and imines were used (please see the reference: Org. Lett. 2022, 24, 5982–5987), although the chelated transition state was proposed totally different in which the chiral induction was through a dynamic kinetic resolution .
